# CREB Is Indispensable to KIT Function in Human Skin Mast Cells—A Positive Feedback Loop between CREB and KIT Orchestrates Skin Mast Cell Fate

**DOI:** 10.3390/cells13010042

**Published:** 2023-12-24

**Authors:** Gürkan Bal, Jean Schneikert, Zhuoran Li, Kristin Franke, Shiva Raj Tripathi, Torsten Zuberbier, Magda Babina

**Affiliations:** 1Fraunhofer Institute for Translational Medicine and Pharmacology ITMP, Immunology and Allergology IA, 12203 Berlin, Germany; guerkan.bal@charite.de (G.B.); jean.schneikert@charite.de (J.S.); zhuoran.li@charite.de (Z.L.); kristin.franke@charite.de (K.F.); shiva-raj.tripathi@charite.de (S.R.T.); torsten.zuberbier@charite.de (T.Z.); 2Institute of Allergology, Charité–Universitätsmedizin Berlin, Corporate Member of Freie Universität Berlin and Humboldt Universität zu Berlin, Hindenburgdamm 30, 12203 Berlin, Germany

**Keywords:** mast cells, skin, transcription factors, CREB, SCF, KIT, survival, apoptosis, cell cycle, proliferation

## Abstract

Skin mast cells (MCs) are critical effector cells in acute allergic reactions, and they contribute to chronic dermatoses like urticaria and atopic and contact dermatitis. KIT represents the cells‘ crucial receptor tyrosine kinase, which orchestrates proliferation, survival, and functional programs throughout the lifespan. cAMP response element binding protein (CREB), an evolutionarily well-conserved transcription factor (TF), regulates multiple cellular programs, but its function in MCs is poorly understood. We recently reported that CREB is an effector of the SCF (Stem Cell Factor)/KIT axis. Here, we ask whether CREB may also act upstream of KIT to orchestrate its functioning. Primary human MCs were isolated from skin and cultured in SCF+IL-4 (Interleukin-4). Pharmacological inhibition (666-15) and RNA interference served to manipulate CREB function. We studied KIT expression using flow cytometry and RT-qPCR, KIT-mediated signaling using immunoblotting, and cell survival using scatterplot and caspase-3 activity. The proliferation and cycle phases were quantified following BrdU incorporation. Transient CREB perturbation resulted in reduced KIT expression. Conversely, microphthalmia transcription factor (MITF) was unnecessary for KIT maintenance. KIT attenuation secondary to CREB was associated with heavily impaired KIT functional outputs, like anti-apoptosis and cell cycle progression. Likewise, KIT-elicited phosphorylation of ERK1/2 (Extracellular Signal-Regulated Kinase 1/2), AKT, and STAT5 (Signal Transducer and Activator of Transcription) was substantially diminished upon CREB inhibition. Surprisingly, the longer-term interference of CREB led to complete cell elimination, in a way surpassing KIT inhibition. Collectively, we reveal CREB as non-redundant in MCs, with its absence being incompatible with skin MCs’ existence. Since SCF/KIT regulates CREB activity and, vice versa, CREB is required for KIT function, a positive feedforward loop between these elements dictates skin MCs’ fate.

## 1. Introduction

The stem cell factor (SCF)/KIT axis represents the most crucial receptor tyrosine kinase (RTK) system of mast cells (MCs), enabling differentiation, proliferation, survival, and functional programs throughout the lifecycle of these cells [1,2,3,4,5].The importance of KIT for the lineage is underlined by the association of somatic gain-of-function mutations (especially the D816V mutation) with systemic mastocytosis [1,4,6]. Various signaling modules are initiated following SCF-mediated KIT-dimerization, encompassing kinase networks and transcription factors (TFs) [5,7,8].

CREB (cAMP response element binding protein) is a versatile TF with over 4000 binding sites in the human genome [9]. The CRE motif (cAMP response element) consists of an 8-nucleotide stretch (TGACGTCA) that is frequently located around 100 bases upstream of the promoter TATA box [10]. As a prototypical stimulus-inducible TF, CREB requires posttranslational modification for activity. The most potent modulation is phosphorylation at Ser-133, which enables transactivation through the recruitment of coactivators to the transcriptional machinery [10,11,12]. We recently reported that CREB experiences efficient phosphorylation following SCF activation of skin MCs [13]. Phosphorylation occurs, at least in part, in an ERK-dependent manner, but is independent of other mitogen-activated protein kinases (MAPKs) or Phosphoinositide 3-kinase (PI3K). CREB has important implications downstream of KIT in MCs, including survival promotion and control of immediate early genes [13]. Thereby, CREB was identified as a critical downstream effector of the SCF/KIT axis operating in MCs of cutaneous origin.

Though MC ontogeny has not been completely resolved, and consists of distinct waves through which peripheral tissues are seeded with MCs during embryonic/fetal development and after birth (at least in the mouse), MCs belong to the myeloid branch of the hematopoietic system [14,15,16]. Terminal differentiation occurs through contact with peripheral tissues and under the influence of local growth factors, where SCF arguably assumes the most significant role [1,17]. While most hematopoietic cells downregulate KIT, as they differentiate towards the distinct lineages, KIT is maintained at high levels during MC development, and is essential for their survival, proliferation, and function.

The maintenance of KIT expression in the MC lineage is intriguing and not completely understood. KIT preservation likely requires a specific constellation of TFs and epigenetic modifications, as well as post-transcriptional processes. Several TFs have been reported to regulate KIT, including Sp1, GATA2, Runt-related transcription factor 1 (RUNX1), Stem cell leukemia (SCL), Lim-only 2 (LMO2), and GATA-1/GATA-2 [18,19,20,21], although the precise binding motifs in the KIT promoter, to which these factors bind, vary between cells, as exemplified by a comparison between hematopoietic stem/progenitor cells and MCs [22]. The post-transcriptional mechanisms involve regulation by micro-RNAs [23], but also by rapid turnover, including internalization and degradation following ligand binding [5,24]. KIT activity is also restrained by several negative regulators, including SOCS6 (suppressor of cytokine signaling 6), and the recently identified CIC (capicua) and RHEX (regulator of human erythroid cell expansion) [8,25,26,27].

In the skin, MCs assume important roles and help coordinate the host’s defenses against invading pathogens, and they can eliminate toxins and venoms [28,29,30,31]. However, when overabundant and/or aberrantly controlled, skin MCs contribute to a host of chronic inflammatory dermatoses, including atopic and contact dermatitis, psoriasis, prurigo, rosacea, urticaria, and mast cell activation syndrome [6,32,33,34,35,36,37,38,39,40,41]. Through the formation of operating units with sensory neurons, skin MCs initiate not only inflammatory circuits but also the sensation of itch [42,43,44,45,46,47,48,49]. Itch is also a common symptom of chronic skin diseases, including atopic dermatitis. By their almost exclusive expression of MRGPRX2 (Mas-related G-protein-coupled receptor member X2) in MCs in the skin and a few other locations, cutaneous MCs are the primary responders to many drugs, as well as endogenous neuro- and host defense peptides [46,49]. Therefore, modulation of MC hyperactivity in disease settings is highly desirable.

The constant presence of KIT signaling is obviously required for the maintenance of the cutaneous MC population, since interference with KIT by means of CDX-0159 (a humanized antibody that inhibits KIT activation by SCF), eradicates MCs [50]. Conversely, an SCF injection into human skin substantially enlarges the MC compartment [51].

As highlighted above, we recently found CREB to be a critical downstream effector of the SCF/KIT network in MCs. Here, we embarked on the reverse process, and analyzed the contribution of CREB to KIT functionality. Surprisingly, we found conclusive evidence that KIT depends on an intact CREB program. CREB maintains KIT surface expression independently of KIT mRNA. Upon CREB interference, the reduction in KIT severely hampers signaling and functional programs elicited via the RTK. This indicates the existence of an unrecognized feedforward loop between KIT and CREB that is effective in MCs, and may be a key contributor to MC-associated pathology.

## 2. Materials and Methods

### 2.1. Cells and Treatments

MCs were isolated from human foreskin tissue as described [52]. Each mast cell preparation/culture originated from several (2–10) donors to achieve sufficient cell numbers, as is routinely performed in our lab [53,54,55,56,57]. The skin was obtained from circumcisions, with the written, informed consent of the patients or their legal guardians, and approval by the university ethics committee (protocol code EA1/204/10, 9 March 2018). The experiments were conducted according to the Declaration of Helsinki principles. Briefly, the skin was cut into strips and treated with dispase (26.5 mL per preparation; activity: 3.8 U/mL; Boehringer-Mannheim, Mannheim, Germany) at 4 °C overnight. The epidermis was removed, and the dermis was finely chopped and digested with 2.29 mg/mL collagenase (activity: 255 U/mg; Worthington, Lakewood, NJ, USA), 0.75 mg/mL hyaluronidase (activity: 1000 U/mg; Sigma, Deisenhofen, Germany), and DNase I at 10 µg/mL (Roche, Basel, Switzerland). Cells were filtered stepwise from the resulting suspension (100 and 40 µm strainers, Fisher Scientific, Berlin, Germany). MC purification was achieved using anti-human c-Kit microbeads (#130-091-332) and an Auto-MACS separation device (both from Miltenyi-Biotec, Bergisch Gladbach, Germany), resulting in 98–100% pure preparations (using acidic toluidine blue staining, 0.1% in 0.5 N HCl (Fisher Scientific), as described in [58,59]).

Purified skin MCs from individual preparations were cultured in Basal Iscove’s medium with 10% FCS (both from Biochrom, Berlin, Germany) in the presence of SCF, and IL-4 freshly provided twice weekly when the cultures were readjusted to 5 × 10^5^/mL. MCs were automatically counted using CASY-TTC (Innovatis/Casy Technology, Reutlingen, Germany) [56,60].

Experiments were performed 3-4 d after the last addition of growth factors. For inhibition studies, cells were pre-incubated with 666-15 (CREB inhibitor; 5 µM unless otherwise stated; from Merck Chemicals, Darmstadt, Germany), Pictilisib (PI3K inhibitor; 10 µM) from Enzo Life Sciences, Germany, or imatinib-mesylate (Gleevec, KIT inhibitor; 10 µM; from Biozol Diagnostica, Eching, Germany), for 30 min, then stimulated (or not) with SCF (100 ng/mL). 666-15 was reported to inhibit the interaction between CREB and its co-activators CREBBP (CREB binding protein) and EP300 (E1A Binding Protein P300), and its potency and selectivity have been shown in previous literature [61,62,63].

### 2.2. Immunoblot Analysis

After pre-treatment with inhibitors for 15 min and/or stimulation with SCF (100 ng/mL), MCs were collected by centrifugation and immediately solubilized in SDS-PAGE (Sodium Dodecyl Sulphate-Polyacrylamide Gel Electrophoresis) sample buffer and boiled for 15 min (whole-cell lysates). Samples with equal cell numbers were subjected to immunoblot analysis. Membrane blocking was performed in 5% (*w*/*v*) low-fat milk powder (Carl Roth, Karlsruhe, Germany) solution for 30 min. The following primary antibodies were purchased from Cell Signaling Technologies (Frankfurt am Main, Germany): anti-p-CREB (S133, #9198), anti-p-ERK1/2 (T202/Y204, #9101), anti-p-STAT5 (Y694, #9359), anti-t-ERK1/2 (#9102), and anti-Cyclophilin B (#43603). For the detection antibody, a goat anti-rabbit IgG peroxidase-conjugated antibody was used (Merck, Darmstadt, Germany #AP132P). For the consecutive development of several molecules on the same membrane, the antibodies (primary and secondary) were removed from the membrane after each detection step with incubation in 0.5 N NaOH (Carl Roth, Karlsruhe, Germany) for 15 min. After each stripping step, the membrane was blocked in 5% (*w*/*v*) low-fat milk powder for 30 min (as above), followed by incubation with the next primary antibody. Proteins were visualized using a chemiluminescence assay (Weststar Ultra 2.0, Cyanagen, Bologna, Italy) according to the manufacturer’s instructions. Bands were recorded on a chemiluminescence imager (Fusion FX7 Spectra, Vilber Lourmat, Eberhardzell, Germany). Semi-quantification of recorded signals was performed using ImageJ software (Rasband, W.S., ImageJ, U. S. National Institutes of Health, Bethesda, MD, USA, https://imagej.nih.gov/ij/ (last accessed on 01 September 2023), 1997–2018). Individual intensity values for the detected proteins were normalized to the intensity of the housekeeping protein cyclophilin B of the same membrane. Cyclophilin B is an ER-specific cyclophilin [64], and is generally used for whole cell/cytosolic lysates. It was chosen mainly due to its size, because it is detected at around 20 kDa and, therefore, does not interfere with any other antibody.

### 2.3. BrdU Incorporation

For proliferation assays, the BrdU Flow Kit (BD Bioscience, Heidelberg, Germany) was utilized, as described [54].

For short-term incubation with CREBi, cells were pretreated with the inhibitor (at 5 µM) or vehicle control for 24 h. CREBi was removed on the first day through two washing steps, and the cells were replated in fresh media supplemented (or not) with BrdU (10 µM) and SCF (100 ng/mL). On the third day, SCF was freshly added to ensure potent cell cycle progression. Cells were harvested on the fifth day and processed for staining. Measurements were performed on a Sony ID7000™ Spectral Cell Analyzer (Weybridge, Surrey, UK).

For long-term treatment with CREBi, MCs (2 days after the last feeding) were plated on day 0 at a density of 5 × 10^5^ cells/mL and incubated in the presence (or absence, as the control) of BrdU (10 µM). The inhibitors Gleevec (5 µM), Pictilisib (10 µM), CREBi (5 µM), or vehicle were added 30 min later. After another 30 min incubation, cells were supplemented with 100 ng/mL SCF. Two days later, the medium was replaced with fresh medium containing the inhibitors and SCF. Cells were harvested on the fifth day and processed as above.

### 2.4. Flow Cytometry

MCs were blocked with human AB serum (Biotest, Dreieich, Germany) for 15 min at 4 °C and then stained with a specific anti-CD117 (Miltenyi-Biotec #130-111-593) antibody for 30 min at 4 °C. Corresponding isotype controls were used in each experiment. After incubation, cells were washed in phosphate-buffered saline (PBS) and resuspended in fluorescence-activated cell sorting (FACS) buffer consisting of 2% fetal bovine serum in PBS. The cells were immediately processed in a Sony ID7000™ Spectral Cell Analyzer. The data were analyzed with FlowJo analysis software Version 10.7 (FlowJo LLC, Ashland, OR, USA).

### 2.5. Caspase-3 Activity

Caspase-3 activity of MCs was detected using a luminometric assay kit (Caspase-Glo 3/7; Promega, Mannheim, Germany), according to the manufacturer’s instructions and as previously described [55]. The assay provides a proluminescent caspase-3/7 substrate, which contains the sequence DEVD that is cleaved to release luminescence. Luminescence detection was performed using a VICTOR X5 2030 Multilabel HTS Microplate Reader (Perkin Elmer, Berlin, Germany) operated with the standard luminescence protocol.

### 2.6. Reverse Transcription-Quantitative PCR (RT-qPCR)

MCs (at 5 × 10^5^ cells/mL) were treated with inhibitors for 15 min prior to SCF addition (100 ng/mL) for 25 min, after which time cells were harvested for RNA extraction. Briefly, RNA was isolated using a NucleoSpin RNA kit from Macherey-Nagel (Düren, Germany) following the manufacturer’s instructions. cDNA synthesis (reverse transcription kit from Fisher Scientific) and RT-qPCR were performed using optimized conditions as described elsewhere [52], using materials from Roche (Roche Diagnostics, Mannheim, Germany). The primer pairs are summarized in Table 1. They were synthesized by TibMolBiol, Berlin, Germany. The 2^−ΔΔCT^ method was used to quantify the relative expression levels of the target genes to three reference genes (appearing at the end of Table 1).

### 2.7. Accell^®^-Mediated RNA Interference

A well-established and efficient siRNA method for skin MCs was utilized [8,54,55,65,66,67,68,69]. In brief, skin MCs were transfected twice (on day 0 and day 1) using CREB-targeting siRNA (E-003619-00-0050, Dharmacon) or MITF-targeting siRNA (E-008674-00-0050, Dharmacon), or control siRNAs [13] (each at 1 µM), for a total of 2 or 3 d in Accell^®^ medium (Dharmacon, Lafayette, CO, USA) (supplemented with Non-Essential Amino Acids and L-Glutamine (both from Carl Roth)). The transfection (2 d) was performed in the presence of SCF to guarantee survival during transfection, before the cells were replated and cultured. After transfection, cells were harvested for downstream applications (flow cytometry, RT-qPCR). Since MITF has a long half-life [70], different transfection times were tested. The best transfection efficiency was determined to be for 3 d.

### 2.8. Statistics

Statistical analyses were carried out using PRISM 8.0 Version 10.1.1 (GraphPad Software, La Jolla, CA, USA). Comparisons between two groups were performed using the paired Student’s *t*-test, with a *p* value of less than 0.05 considered statistically significant. For comparisons across more than two groups, an RM one-way ANOVA with Dunnett’s multiple comparisons test was used. A one-sample *t*-test was applied to assess the significance of normalized values.

## 3. Results

### 3.1. CREB Is Essential to KIT Expression in Skin MCs

KIT expression on MCs is very prominent, and also applies to MCs of human skin origin [8,24,71]. Moreover, the SCF/KIT axis maintains an intact skin MC compartment in vitro [60,72,73], as well as in vivo [74]. Conversely, CREB is a downstream component of KIT’s pro-survival machinery, as recently uncovered [13].

To analyze the inverse scenario, i.e., the involvement of CREB in KIT expression, cells were treated with the CREB-selective inhibitor (CREBi) versus vehicle control for 2 d. As shown in Figure 1a, we observed substantial downregulation of KIT by CREBi. The MCs were still alive, but were at the beginning of losing their viability because of the downregulation of KIT expression. Therefore, we also observed an altered peak pattern upon CREBi treatment (KIT^low^ and KIT^high^ populations, Figure 1a). In fact, we recently reported that CREB activity is important for ensuring MC survival (without information on its relationship to KIT) [13], which explains the more pronounced KIT^low^ population after interference with CREB. In a parallel setting, SCF was freshly added 4 h after CREBi. On the one hand, SCF maintains skin MC survival [8,60,73]; on the other, SCF also leads to the rapid internalization of its receptor [24,56] (Figure 1b). Both processes are well illustrated in the overlays of Figure 1a versus Figure 1b. Notwithstanding this, KIT reduction upon CREB suppression was similarly discernible independently of whether SCF was freshly provided (Figure 1a,b).

To establish the order of events, it was important to clarify whether a restrained KIT expression is secondary to the initiation of the beginning of viability loss elicited by the same substance, or whether it also occurs when the cells are still fully unperturbed. We, therefore, tested at an earlier point. The cells exposed to CREBi for 24 h maintained their KIT distribution, and only displayed one peak in the respective histogram (Figure 1c,d). Yet, they showed a vast reduction in KIT, indicating that downregulation is not a bystander to the slowly commencing cell death, but that KIT loss comes first. In fact, this was a hint of CREBi’s pro-apoptotic role, which became fully evident later, and is brought about through the reduction in KIT. The altered surface expression was not accompanied by changes in KIT mRNA, indicating that downregulation of the receptor was unrelated to the activity of its gene. It is well documented that KIT protein expression is subject to post-transcriptional regulation of various sorts, and does not correlate to KIT transcript abundance (or even shows, instead, an inverse correlation [24,71]).

To verify the above, we used CREB-selective siRNA to curtail CREB via RNA interference [13,66]. The siRNA resulted in a reduction of approximately 50%, as in our previous report (Figure 2a) [13]. This was sufficient to drive KIT expression down, and it was also strongly diminished in the knockdown setting (Figure 2b). The results for both strategies were, therefore, comparable. Again, the downregulation of KIT protein was not driven by an altered transcript abundance (Figure 2c). The results of both strategies combined establish that KIT expression requires an unperturbed CREB system.

Since the microphthalmia factor (MITF) has been described as regulating KIT expression (mainly in murine MCs and human gastrointestinal tumors [75,76,77]), we finally compared CREB to MITF. To our surprise, RNA interference with (NO ARTICLE) MITF had no impact on KIT surface or KIT mRNA expression (Figure 2d–f), further highlighting the selectivity of CREB.

### 3.2. CREB Maintains KIT Functionality–Pretreatment with CREB Inhibitor Compromises KIT Signaling

As with the MC lineage in general, KIT is also expressed at very high levels in skin-derived MCs. It was, therefore, important to learn whether the attenuated expression due to CREB perturbation has functional consequences. Therefore, the signaling capacity of KIT was explored upon CREBi pre-exposure [8], which reduced KIT surface expression, as shown above. To this end, MCs were pretreated with CREBi for 1 d, washed extensively, and stimulated with SCF for different times.

Figure 3 demonstrates the crucial modules activated downstream of KIT in skin MCs according to our recent publication [8], namely, ERK1/2, PI3K/AKT, and STAT5. Also in this study, ERK phosphorylation was most strongly induced by SCF (over the baseline), with the maximum at 5 min, and then steadily declining in the control setting (Figure 3a, green bars). When KIT expression was reduced with a 24 h pre-treatment with CREBi, the time-course was largely maintained, but the signals were reduced at each point of SCF stimulation (Figure 3a, blue bars; Figure 3d), suggesting the diminished efficiency of KIT to initiate signaling. In contrast, a five-minute pre-incubation with CREBi had no impact on the phosphorylation of the signaling components (data not depicted), suggesting that the effects of CREBi were indirect and required KIT as an intermediary. The results for pAKT were similar, both regarding the kinetics and the effects of CREBi (Figure 3b,d). The phosphorylation of STAT5 occurred more slowly compared to that of ERK and AKT (Figure 3c,d), yet the inhibitory effect of CREBi was at least as potent as that for the kinases. Collectively, the reduction in KIT at the cell surface results in an attenuated capacity to elicit signal transduction upon SCF-mediated dimerization. This indicates that, despite the vast expression of this RTK on MCs, its reduction following CREB interference has functional consequences.

### 3.3. CREB Is Essential for KIT-Dependent Anti-Apoptosis and Cell Cycle Progression

Having found that KIT expression requires unperturbed CREB activity, we further explored whether SCF-mediated MC survival is affected by the preceding CREB inhibition. To this end, the CREBi and vehicle-pretreated cells were washed extensively, and the cells were exposed to freshly provided SCF (or kept in the absence of SCF) for 2 days. Apoptosis was measured using caspase-3 activity, as it was the most sensitive and quantitative method among a number of assays tested [55].

As expected, SCF potently interfered with MC apoptosis (Figure 4a). Pretreatment with CREBi for 24 h dampened KIT functionality and attenuated SCF’s anti-apoptotic potential (Figure 4b, two columns on the left). CREBi pretreatment had, however, no effect in the absence of SCF (Figure 4b, two columns on the right), indicating that dampened survival required the SCF/KIT axis.

In addition to the extensive washing, we implemented a further strategy to exclude the possibility that residual CREBi activity is responsible for the effects (pro-apoptotic action, in this case), rather than it being the result of KIT reduction. Therefore, the 24 h supernatants from CREBi- and vehicle-treated groups were collected. MCs were treated with these supernatants side-by-side against a control treated with regular medium. Interestingly, the supernatants contained (NOT displayed) pro-apoptotic activities against naïve cells, which may encompass MC waste products (Figure 4c). Importantly, however, there was no difference between the supernatants collected from CREBi-exposed or vehicle-exposed MCs, excluding the presence of meaningful spurs of CREBi. We conclude that CREB inhibition in this setting operates by reducing KIT, which then dampens SCF-dependent anti-apoptosis.

In the above experiments, the anti-apoptotic effect of SCF and its reduction following prior treatment with CREBi was determined after 2 days. We were also interested in whether KIT attenuation (elicited by transient CREBi) can have longer-term effects. This was tested in KIT-dependent cell cycle progression elicited by repetitive doses of SCF over a 5 day period [8,54,58]. As in the above experiments, CREBi was applied only once for 24 h, and was then removed from the system. KIT downregulation was confirmed using flow cytometry. The MCs were then incubated with fresh medium containing 100 ng/mL SCF in the presence (or absence) of BrdU. SCF (at the same concentration) was re-added 3 d later. On day 5, the cells were harvested for inspection of BrdU incorporation.

Surprisingly, the one-time addition of CREBi was sufficient to interfere with KIT-dependent proliferation. In fact, the cells that had incorporated BrdU dropped from 42.9% to 25.9% on average (Figure 5a,c). These numbers are very similar to the proportion of cells that are in (without involved, replace by “that are” if required) the process of synthesizing DNA, i.e., 41.7% in the vehicle-pretreated versus 25.5% in the CREBi-pretreated cells (Figure 5a,b). Correspondingly, the proportion of G0/G1 cells rose from 52.1 to 65.7% for the mean of all experiments (Figure 5a,b). The proportion of apoptotic cells was generally low in both groups, with higher values (around ≈ 20%) reached in only two out of the five experiments (Figure 5a). This indicates that the cells were mostly protected from apoptosis due to a sufficient provision of SCF, while the apoptotic cells that appeared earlier had mostly been eliminated by day 5 in this setting. Collectively, despite its vast expression, the precise expression level of KIT dictates survival prolongation and proliferative fitness of skin MCs mediated by its ligand. The perturbation of KIT, therefore, manifests as a reduction in the sensitivity to SCF.

### 3.4. Prolonged Inhibition of CREB Leads to MC Elimination, in a Way Exceeding the Inhibition of KIT Itself

CREB is downstream of KIT, experiencing activation and organizing SCF-mediated survival, as reported recently [13]. The preceding paragraphs indicated that CREB also acts upstream of KIT, by stabilizing its expression and, thereby, maintaining its function. Since persistent KIT activation is indispensable for MCs to enter the cell cycle, we presumed that the suppression of CREB may have similar effects as the suppression of KIT itself. To test this, CREBi and KITi (imatinib mesylate, a potent inhibitor of the KIT tyrosine kinase [78]) were used side-by-side. Moreover, Pictilisib (a PI3K inhibitor) was employed as a further positive control [8].

Surprisingly, interference with CREB over a five-day period resulted in almost complete cell demise. This was already detectable in the scatter plots, where few identifiable cells were detectable; in fact, only 2–25% of events were still within the cell gate, while the majority was part of the cell debris fraction (Figure 6a,d). Moreover, of the few identifiable cells, ≈92% were apoptotic (Figure 6b and Appendix A). Therefore, a compromised CREB function is incompatible with skin MC life. For KITi and PI3Ki, substantially more events compared to CREBi were still identifiable in the cell gate (18–86% and 29–92%, respectively, Figure 6a,d). Of these, around 66% and 40% were apoptotic (Figure 6b and Appendix A). As expected, BrdU incorporation was very low, with 6.5%, 6.8%, and 12.9% for CREBi, KITi, and PI3Ki, respectively, compared to >50% in the control (Figure 6c,e), substantiating that their targets are vital for MC proliferation. This corresponds with low proportions of S-phase cells (Appendix A). Notably, the global effect on cell recovery and BrdU incorporation was even more accentuated for CREBi than for KITi. This suggests that CREB does not only sustain MCs via KIT, but also affects KIT-independent survival mechanisms, though the KIT-dependent portion is probably dominant. Since KIT causes CREB activation [13], while CREB maintains KIT function, we hereby reveal a self-perpetuating loop between these elements. Through mutual potentiation, the loop becomes self-sustaining, as long as a sufficient SCF provision is guaranteed.

## 4. Discussion

KIT is a crucial RTK of the hematopoietic system. While vital at the level of hematopoietic stem and progenitor cells, most lineages downregulate KIT as they differentiate towards their mature endpoints. MCs are an exception, as they continue to express and even further upregulate KIT in their fully differentiated forms [72,79,80]. The SCF/KIT axis maintains survival, regulates the phenotype and metabolism of the cells, and enables optimal responses to antigens. In the skin micro-milieu, SCF is expressed by structural cells, such as keratinocytes [81,82], endothelial cells [83,84], fibroblasts [85,86], and dermal papilla cells [17,87]. MCs can be isolated from the lung, gut, and skin and, at high concentrations, SCF prompts these mature MCs to re-enter the cell cycle [54,60,88,89,90,91,92,93,94,95,96,97]. The modifications elicited by SCF in skin MCs are accompanied by massive changes in the phosphoproteome, with roughly 5400 out of 10,500 phosphosites incurring changes in abundance [8].

Although CREB is an evolutionarily ancient TF, that has been extensively studied across tissues and cells of primitive species and humans alike [98,99], its role in the hematopoietic system is underexamined compared to its role in the nervous system. Notwithstanding, several studies have found that CREB overexpression is associated with the uncontrolled proliferation and survival of hematopoietic progenitors and leukemia cells, especially of myeloid origin, the branch to which MCs also belong [10,100,101]. No connection has been made between CREB and KIT, however.

It is noteworthy that MCs were among the highest expressors of CREB in FANTOM5 (Functional Annotation of the Mammalian Genome) [72,79,80]. Its association partners, CREBBP and EP300, were likewise preferentially expressed in the lineage [102]. However, despite extensive research into CREB in various tissues and cells, its function in MCs is surprisingly ill defined. In the few available studies, CREB has been associated with the production of inflammatory mediators elicited by different receptors, including FcεRI, where it acted as a positive (and sometimes a negative) regulator [103,104,105,106]. Conversely, apart from our recent study [13], which provided the basis of the current investigation, CREB function in lineage specification and maintenance of mature MCs has not been examined. We here uncovered CREB as an organizer of MC fate, whereby the TF acts high up in the hierarchy, regulating central programs like survival and proliferation. This is achieved, at least to a significant degree, by maintaining the functionality of the major receptor tyrosine kinase operative in these cells, namely KIT, where KIT expression itself is under positive regulation from CREB. While KIT functionality depends on unperturbed CREB in skin MCs, the dependence is not transcriptional, since reduced receptor expression was accompanied by stable KIT mRNA levels. This largely excludes the possibility that CREB operates through the direct activation of the KIT gene. It is well established that KIT is extensively regulated by post-transcriptional mechanisms. A reflection of this is the lacking correlation between *KIT* mRNA and KIT protein expression across human subjects [71]. An important regulatory mechanism is KIT’s rapid internalization following SCF-triggered dimerization [24]. Other stimulatory cues can lead to the shedding of KIT, a mechanism also found in skin MCs [24]. Likewise, KIT is very sensitive to inhibitors of translation, suggesting rapid turnover with the need for constant resynthesis [24]. Since KIT downregulation occurs as an indirect process, it may, therefore, be assumed that CREB itself or its downstream TFs (like FOS, JUNB, and NR4A [13]) either increase activators (or repress inhibitors) of KIT translation or transport. Alternatively, CREB or its downstream TFs may repress the genes involved in KIT degradation, thereby maintaining KIT on the surface for longer periods. In fact, CREB has been shown to also inhibit several genes through the formation of repressive heterodimers with other CREB family members [107]. Downstream TFs, like the NR4A subfamily members, can likewise act as repressors of gene expression [108]. As mentioned in the introduction, CREB can target up to 4000 genes; the identification of the most significant targets involved in the regulation of KIT will require a combination of global omics efforts along with the selective manipulation of CREB function. This information could guide researchers about the best way to target aberrant pathways in KIT-associated diseases, such as mastocytosis and cancer. We are currently planning follow-up studies to pinpoint the processes affected by CREBi (translation, turnover, or shedding) and the potential genetic events driving them.

In contrast to CREB, other TFs have been more tightly associated with MC development and function, in particular, MITF, GATA-2, GATA-1 and, to a certain degree, PU.1 [17,109,110,111,112,113]. While most of the studies were on rodent MCs, these factors are also highly expressed by human skin MCs [72,114], and some recent evidence has established key concepts for how they impact functional aspects of MCs in humans [70,115]. It, therefore, came as a surprise that MITF perturbation does not interfere with KIT expression in skin MCs (either RNA or protein). In previous studies, MITF was reported to positively regulate KIT, although mainly in murine MCs [75]. The lack of regulation by MITF in human skin MCs emphasizes the differences between murine and human MCs, and further underlines the significance of CREB in KIT maintenance.

Our study demonstrates that the consequences of a CREB shutdown in skin MCs are drastic, exceeding interference with well-known pathways. We recently reported that ERK and PI3K activities downstream of KIT mediate anti-apoptosis in a partially redundant manner in the short run [8]. This mutual substitution can explain why the inhibition of PI3K does not reach the same level of interference as the inhibition of KIT by imatinib mesylate, with the latter blocking the PI3K, the ERK1/2, the STAT5, and other modules of the axis [8]. Unexpectedly, interference with CREB had more serious consequences than interference with KIT. This was already found at the level of identifiable cells recovered, that are shown in the FCS/SSC plots, but even within the low proportion of cells recovered, almost all were apoptotic after long-term CREBi. Surprisingly, while also strongly affected by KIT inhibition, the consequences at the level of surviving cells were less pronounced. It is thus probable that MCs produce autocrine survival factors (other than SCF) that likewise interfere to some degree with apoptosis in a KIT-independent manner. This was also hinted at in our previous study [13]. If CREB was required for (the major) KIT-dependent and (the minor) KIT-independent survival promotion, its suppression would lead to the full cell demise observed.

The survival of skin MCs requires mechanisms that encompass Mcl-1 and Bcl-xl as the key downstream effectors [55,65]. Mcl-1 is regulated by STAT5, while Bcl-xl requires JNK function, leading to the activation of (an) as yet to be uncovered TF(s) [65]. Of note, Mcl-1 is not directly affected by CREB inhibition, but is greatly affected by KIT inhibition [13]. However, through the attenuation of KIT function, the inhibition of CREB will also target Mcl-1 in a timely shifted manner. In fact, STAT5 activation is greatly reduced following CREBi (Figure 3). Of note, the downstream targets of CREB, in particular NR4A2 [13], have been demonstrated to positively regulate neutrophil lifespan [116]. MCs share several features with neutrophils, including their dependence on Mcl-1 [55,117].

In our previous report, we demonstrated that KIT activates CREB [13]. In the current study, we show that CREB maintains KIT. In fact, KIT surprisingly requires constant CREB activity; in its absence, i.e., following RNAi or pharmacological interference, KIT function is substantially attenuated, resulting in severely hampered downstream programs, like survival and proliferation. Viewing the findings from both reports in aggregate, the most striking outcome is the existence of a feedforward stimulatory loop between these two elements. In fact, halting CREB activity has as dramatic or even more dramatic effects than the suppression of KIT itself. Collectively, we identify CREB as a non-redundant factor of MCs, whose dysfunction is incompatible with skin MC existence.

It will be of the utmost interest to explore whether CREB is hyperactive or otherwise altered in mastocytosis and other diseases characterized by increased MC abundance, as well as the impact of the D816V mutation on the loop. Several labs have embarked on the generation of MCs from induced pluripotent stem cells, which allows for the culturing of patient-specific cells [118,119]. While the global inhibition of CREB is predicted to bring about unacceptable levels of toxicity and side-effects due to the factor’s body-wide activity, other components of this pathway, which are more limited to selected cells only, could be envisaged as targets [101]. Along the same lines, the targeting of cell-specific CREB-regulated genes may also be a way forward in MC-mediated or MC-associated diseases, as has been suggested for neuropsychiatric conditions.

## 5. Conclusions

Our study identifies the ancient TF CREB as essential to primary skin MCs. In this regard, CREB operates not only downstream of KIT through ERK-dependent phosphorylation, as shown recently [13], but also upstream by signaling back to maintain the RTK in a functionally active form. Interference with CREB leads to the loss of KIT expression, KIT signaling, and KIT-mediated anti-apoptosis and proliferation. The long-term interference with CREB activity erases skin MCs completely. Thus, a lack of CREB is incompatible with skin MC existence. Interestingly, CREB’s relevance exceeds that of KIT itself, suggesting that it orchestrates both KIT-dependent and KIT-independent survival programs. Collectively, CREB is more important to MC biology than hitherto suspected, especially when partnered up with MCs’ most significant receptor tyrosine kinase.

## Figures and Tables

**Figure 1 cells-13-00042-f001:**
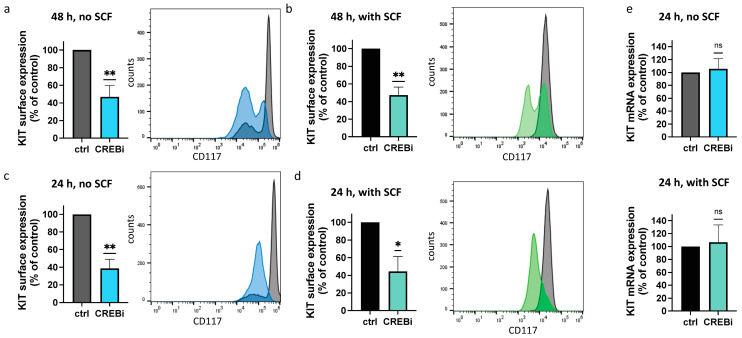
KIT surface expression requires intact CREB activity. MCs were kept in the presence of CREBi or DMSO (ctrl) for the indicated periods, either without or with the addition of SCF. (**a**–**d**) Cells were harvested and incubated with an anti-CD117 antibody for the detection of KIT on the cell surface. The right panels represent the flow cytometry analysis, showing the distribution of KIT in the cell population. The left panels show the quantification of the mean fluorescence intensity relative to the vehicle control, set as 100%, (**a**) at 48 h, (**b**) at 48 h in the presence of SCF, (**c**) at 24 h, and (**d**) at 24 h in the presence of SCF. Shown are the means ± SEM of 4–6 independent experiments. (**e**) Top and bottom: cells were harvested and processed for RT-qPCR. The relative expression of KIT mRNA was normalized to that of the control. The mean ± SEM of 4–7 independent experiments are shown. * *p* < 0.05, ** *p* < 0.01; ns: not significant.

**Figure 2 cells-13-00042-f002:**
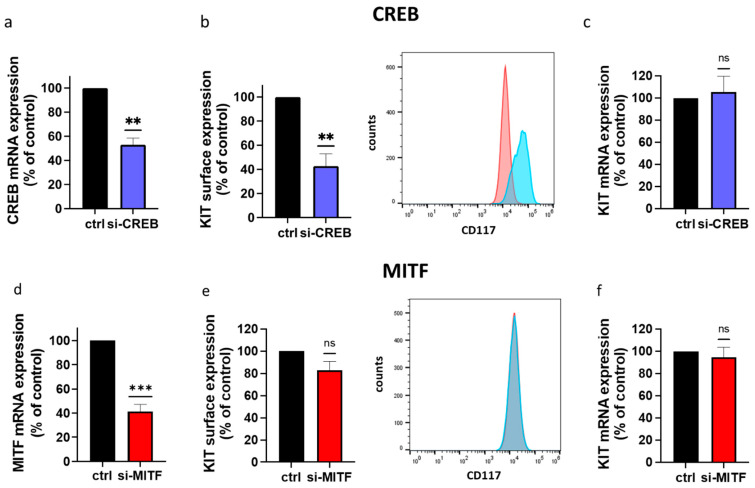
RNA interference with CREB reduces cell surface expression of KIT, while MITF has no significant effect. (**a**–**c**) MCs were transfected with either a control siRNA (ctrl: black) or an siRNA-targeting CREB (si-CREB: dark blue). Cells were kept in the presence of SCF during transfection to ensure survival under the minimal medium NOT AMOUNT! Minimal medium means it contains no serum, or growth factors (Accell) required for the procedure. Cells were harvested 48 h after the initial transfection. (**a**,**c**) RT-qPCR of CREB and KIT mRNAs, respectively. Shown are the means ± SEM of five independent experiments (ns: not significant). (**b**) The right panels are the representative flow cytometry analysis showing the distribution of KIT in the cell population (blue: control siRNA; red: siCREB). The left panels are the quantification of the mean fluorescence intensity relative to the control, set at 100%. Shown are the means ± SEM of five independent experiments. (**d**–**f**) Skin MCs were transiently transfected with either control siRNA (black: ctrl) or MITF-selective siRNA (red: si-MITF) as in (**a**–**c**). Cells were harvested after 72 h (due to the longer half-life of MITF). RT-qPCR and flow cytometry were performed as above. (**d**–**f**) are analogous to (**a**–**c**). Shown are the means ± SEM of 4–6 independent experiments. ** *p* < 0.01, *** *p* < 0.001, with one sample *t*-test; ns: not significant.

**Figure 3 cells-13-00042-f003:**
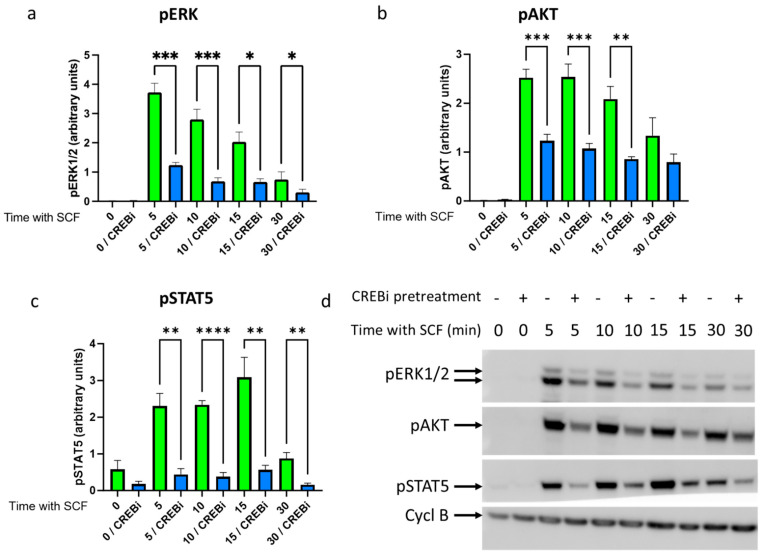
CREB is required for SCF-induced signaling through ERK, AKT, and STAT5. MCs were pretreated with CREBi (blue) or DMSO (green) for 1 day, washed extensively, and stimulated with SCF for the indicated periods in minutes. Cells were harvested and subjected to Western blotting analysis using antibodies against phospho-ERK1/2, phospho-AKT, and phospho-STAT5. An antibody against Cyclophilin B (Cycl B) was included as a loading control and was used for normalization. (**a**–**c**) Relative quantifications of the Western blot signals. Shown are the means ± SEM of four independent experiments. * *p* < 0.05, ** *p* < 0.01, *** *p* < 0.001, **** *p* < 0.0001. (**d**) Representative Western blot experiment, using the indicated antibodies.

**Figure 4 cells-13-00042-f004:**
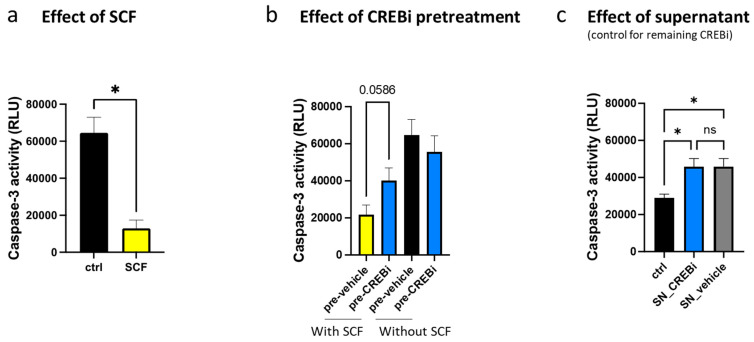
The reduction in KIT following CREB inhibition is required for the induction of apoptosis. (**a**) MCs were deprived of SCF (ctr, black) or kept in the presence of SCF (yellow) for two days. Cells were harvested and the enzymatic activity of caspase-3 was measured. Shown are relative units that are the means of 4 independent experiments ± SEM. (**b**) MC cells were pretreated with either CREBi (pre-CREBi) or DMSO (pre-vehicle) for 1 day, washed extensively, and incubated either in the absence or the presence of SCF for two days, after which the caspase-3 activity was determined. Shown are relative units that are the means of at least 4 independent experiments ± SEM. (**c**) The 24 h supernatants of CREBi (SN_CREBi)- and vehicle (SN_vehicle)-treated cells (in the presence of SCF) were applied to MCs for two days, after which caspase-3 activity was measured. ctrl: control cells (left untreated). Shown are relative units that are the means ± SEM of 4 independent experiments. * *p* < 0.05. ns: not significant.

**Figure 5 cells-13-00042-f005:**
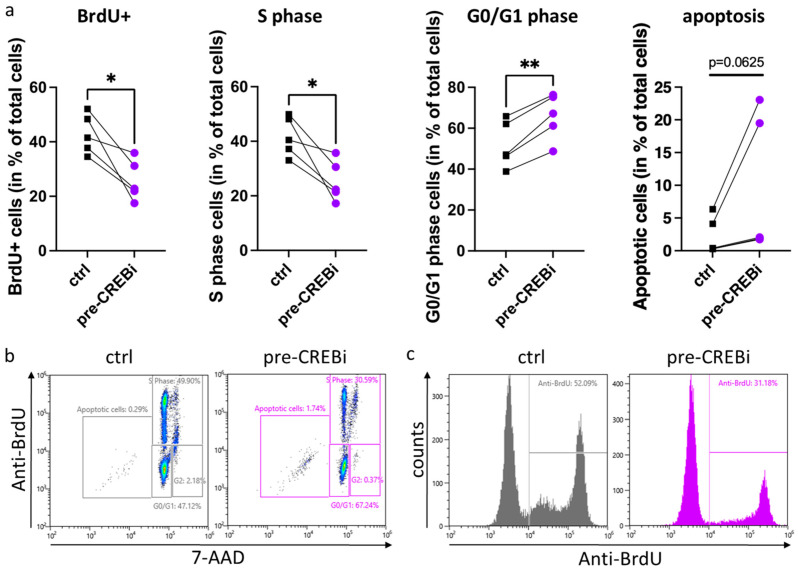
Reduced SCF-triggered proliferation as a result of impaired KIT expression due to CREB inhibition. MCs were treated with either CREBi (pre-CREBi) or DMSO (ctrl) for 24 h, washed extensively, and incubated in the presence of 100 ng/mL SCF and BrdU for 5 days. After application of the anti-BrdU antibody, total DNA was labeled with the 7-AAD dye and cells were analyzed using flow cytometry. (**a**) Effect of CREBi on the proportions of S-phase, G0/G1, and apoptotic cells. (**b**) Representative scatter plots showing the different cell cycle stages based on BrdU fluorescence against the total amount of DNA, either in the vehicle control (left panel) or in the presence of CREBi (right panel). (**c**) Representative histograms, displaying the extent of BrdU incorporation in the cell population upon CREBi treatment (right panel) or in the vehicle control (left panel). n = 5; * *p* < 0.05, ** *p* < 0.01.

**Figure 6 cells-13-00042-f006:**
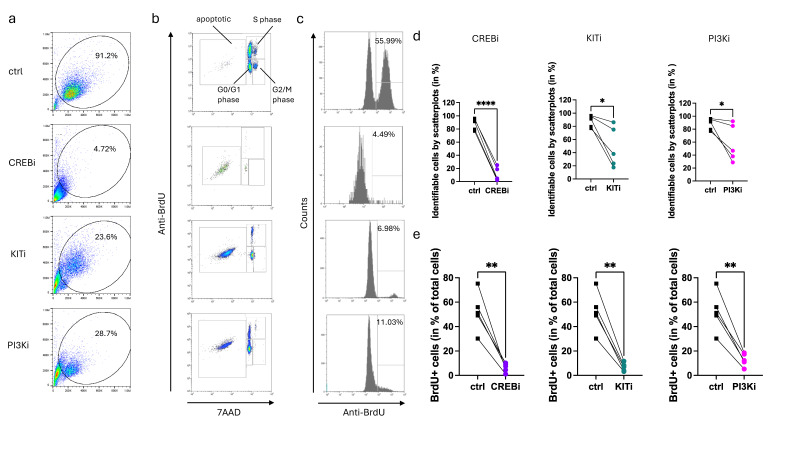
CREBi eliminates skin MCs. MC cells were treated with either CREBi, KITi, PI3Ki or DMSO (ctrl), as indicated, and incubated in the presence of 100 ng/mL SCF and BrdU for a total of 5 days, as detailed in the Methods. Cells were harvested and analyzed using flow cytometry for the incorporation of BrdU and DNA quantity as shown in Figure 5. (**a**) Representative FSC/SSC scatter plots in the presence of the indicated inhibitors. (**b**) Representative scatter plots showing BrdU fluorescence against the total amount of DNA. (**c**) Representative histograms displaying the extent of BrdU incorporation into the cell population upon treatment with the inhibitors. (**d**,**e**) Effect of the indicated inhibitors on the proportions of identifiable cells and on the proportions of BrdU positive cells, respectively. n = 5; * *p* < 0.05, ** *p* < 0.003, **** *p* < 0.0001.

**Table 1 cells-13-00042-t001:** Primer pairs used for RT-PCR.

Gene	Forward 5′-3′	Reverse 5′-3′
*CREB1*	GAGAAGCGGAGTGTTGGTGA	TCCGTCACTGCTTTCGTTCA
*MITF*	CCCTTATTCCATCCACGGGTCTC	ATACTGCTCCTCCGGCTGCTTGT
*HPRT*	GCCTCCCATCTCCTTCATCA	CCTGGCGTCGTGATTAGTGA
*PPIB* *	AAGATGTCCCTGTGCCCTAC	ATGGCAAGCATGTGGTGTTT
*GAPDH*	ATCTCGCTCCTGGAAGATGG	AGGTCGGAGTCAACGGATTT

* The PPIB gene encodes Cyclophilin B.

## Data Availability

No datasets were generated or analyzed during this study. Data are contained within the article and Appendix A.

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
