# Peer review of "CREB Is Indispensable to KIT Function in Human Skin Mast Cells—A Positive Feedback Loop between CREB and KIT Orchestrates Skin Mast Cell Fate"

_cells, 2023, doi:10.3390/cells13010042_

Round 1
Reviewer 1 Report
Comments and Suggestions for Authors
Babina and co-workers have previously established that CREB is activated downstream of KIT ligation in mast cells. In this work, they have investigated the opposite scenario, i.e., whether CREB also acts upstream of KIT. Indeed, they show that CREB suppression, either by siRNA or by using a CREB inhibitor has an inhibitory impact on cell surface KIT expression in primary human skin mast cells. As expected, the suppression of CREB cell surface expression resulted in blunted signalling responses downstream of stem cell factor stimulation of the mast cells, encompassing the ERK, AKT and STAT5 pathways. The authors also demonstrate that CREB inhibition results in increased caspase-3 activity in the mast cells, implying a higher extent of apoptosis. Again, this expected considering the important pro-survival function of the KIT-stem cell factor axis. It is also shown that CREB inhibition results in decreased proliferation and that long-term treatment with CREB inhibitor results in extensive depletion of mast cells. This is an interesting study that increases our knowledge of how KIT is regulated in mast cells. The experiments are well done and are well explained.
The authors could consider some points
1. In paragraph 3.1 it is stated that RNA interference with MITF, in contrast to interference with CREB, had no impact on cell surface KIT expression. This is interesting and also demonstrates the specificity of the anti-CREB approach. I would suggest that the data on MITF are included in Fig1, rather than being placed as a supplementary figure
2. It is intriguing that CREB inhibition suppresses cell surface expression of KIT without affecting the corresponding mRNA levels. The authors discuss the possible background to this in the Discussion, but it would obviously be of interest to provide some insight into this issue, either in this study or in follow up work. For example, does CREB inhibition result in increased KIT internalization, degradation or shedding? This could be experimentally addressed by fluorescence microscopy and Western blot approaches.
3. Second paragraph under 3.3.: The last sentence in this paragraph is somewhat confusingly written, and could be clarified. Also, the data presentation on Fig 4 could be improved. In its present form, it is somewhat confusing that the y-axes scales are different among the panels etc. It is also unclear why panel b is split into two displays?
4. For clarity, I would suggest that the 4 panels in Fig 5b have labels on top, so that it is clear what the left/right panels represent.
Comments on the Quality of English Language
none
Author Response
The reply is in the uploaded PDF document

Reviewer 2 Report
Comments and Suggestions for Authors
The authors describe that CREB regulates the expression of c-Kit on human mast cells.
This is a well written a very interesting manuscript, since this is the first time that CREB is described as an upstream regulator of c-Kit on mast cells.
Experimentally, there are no open questions and the results are clearly presented and show that CREB mediates the surface expression of c-Kit and the functionality of the downstream signalling and effector functions.
The author did a great job! Congratulation!
Author Response
We thank the reviewer very much for their brief summary, praise, and highly encouraging comments. They will help us to design follow-up studies to delve deeper down into the mechanisms driving these changes. We have made some minor edits based on the suggestions from the other reviewers.
Reviewer 3 Report
Comments and Suggestions for Authors
In this manuscript, Bal and colleagues investigated in vitro the role of CREB in KIT signaling in mast cell by using siRNA or chemical inhibitor of CREB. The authors described a positive loop between CREB and KIT and concluded that CREB is fundamental for skin mast cell survival.
The topic is certainly interesting given the complex role of mast cells in the maintenance of skin tissue homeostasis and diseases.
The authors show experience in the field as documented by their previous publications. Indeed, they published similar data on Int J Mol Sci 2023 (DOI: 10.3390/ijms24044135) and the actual manuscript seems its natural extension. I believe that the work does not present great novelty and is based only on in vitro experiments, however it contributes to better understanding an aspect of the complex biology of mast cells.
Major point:
Can the authors add some more information on the mechanism of action of the CREB inhibitor (666-15) used? I wonder if the experiments described in figure 3 are sufficient to demonstrated a direct effect of CREB inhibition on SCF-mediated response or rather whether they are only indicative of a general impairment of the ckit signaling pathway. Are the authors sure that the inhibitor actually blocks CREB?
Minor points:
-figure 1: flow cytometry graphs in panels a-b-c-d lack y-axis titles
-figure 3: correct the y-axis titles (all y-axis report the same wording pERK1/2 even when they refer to pAKT or pSTAT5)
-figure 4: please uniform the scale of y-axis in all the 4 histograms for a better comparison
-methods: please specify the composition of culture media used for skin MC culture (RPMI, DMEM..?).
-methods: In 2.2. paragraph it is reported that MC were stimulated with IL-33… I assume this is a misprint because IL-33 was never mentioned in the manuscript Please correct.
I strongly recommend to improve quality of the submitted figures
Author Response
The reply is in the uploaded pdf document
